# Evaluation of Native Entomopathogenic Fungi for the Control of Fall Armyworm (*Spodoptera frugiperda*) in Thailand: A Sustainable Way for Eco-Friendly Agriculture

**DOI:** 10.3390/jof7121073

**Published:** 2021-12-13

**Authors:** Julius Rajula, Sarayut Pittarate, Nakarin Suwannarach, Jaturong Kumla, Aneta A. Ptaszynska, Malee Thungrabeab, Supamit Mekchay, Patcharin Krutmuang

**Affiliations:** 1Department of Entomology and Plant Pathology, Faculty of Agriculture, Chiang Mai University, Chiang Mai 50200, Thailand; julyraju12@gmail.com (J.R.); B.pittarate@gmail.com (S.P.); 2Research Center of Microbial Diversity and Sustainable Utilization, Faculty of Science, Chiang Mai University, Chiang Mai 50200, Thailand; suwan.462@gmail.com (N.S.); Jaturong_yai@hotmail.com (J.K.); 3Department of Biology, Faculty of Science, Chiang Mai University, Chiang Mai 50200, Thailand; 4Department of Immunobiology, Faculty of Biology and Biotechnology, Institute of Biological Sciences, Maria Curie-Skłodowska University, Akademicka Str. 19, 20-033 Lublin, Poland; aneta.ptaszynska@poczta.umcs.lublin.pl; 5Agricultural Technology Research Institute, Rajamangala University of Technology Lanna, Lampang 52000, Thailand; mthungrabeab@rmutl.ac.th; 6Department of Animal and Aquatic Sciences, Faculty of Agriculture, Chiang Mai University, Chiang Mai 50200, Thailand; supamit.m@cmu.ac.th; 7Innovative Agriculture Research Center, Faculty of Agriculture, Chiang Mai University, Chiang Mai 50200, Thailand

**Keywords:** *Spodoptera frugiperda*, *Beauveria bassiana*, multi-gene, *GAS1* gene, efficacy

## Abstract

Fall armyworm, *Spodoptera frugiperda*, entered Thailand in late 2018 and has now spread in several regions, with devastating effects in maize and rice production, which are some of the most important cereals in the world. Since then, farmers have utilized the available chemical insecticides to try to control it, but their efforts have been futile. Instead, they have ended up using extraordinary dosages, hence threatening non-target species and other fauna and flora, as well as being costly. In this regard, research has been ongoing, aiming to come up with eco-friendly solutions for this insect. We surveyed and collected various isolates of native entomopathogenic fungi intending to test their efficacy against fall armyworm. Six isolates of entomopathogenic fungi were obtained and identified to *Beauveria bassiana* based on morphological characteristics and multi-gene phylogenetic analyses. Thereafter, the six isolates of *B. bassiana* were used to perform efficacy experiments against fall armyworm. Additionally, the glycosyl transferase-like protein 1 (*GAS1*) gene was analyzed. Consequently, all the isolates showed efficacy against *S. frugiperda,* with isolate BCMU6 causing up to 91.67% mortality. Further, molecular analysis revealed that all the isolates possess the *GAS1* gene, which contributed to their virulence against the insect. This is the first report of utilizing native entomopathogenic *B. bassiana* to manage *S. frugiperda* in Thailand, with the revelation of *GAS1* as a factor in inducing virulence and cuticle penetration. This study has provided valuable information on the potential development of *Beauveria bassiana* as an eco-friendly bioinsecticide for the management of fall armyworm in Thailand.

## 1. Introduction

Maize ranks among the topmost commercially produced and utilized cereals globally. Firstly, it is used as human and animal feed. Additionally, maize is used in the food and beverage, paper, bioplastics, pharmaceutical, and textile industries, among others [1,2]. The agricultural sector in Thailand contributes about 9.9% to the Gross Domestic Product (GDP) in Thailand, and maize is among the top five crops on commercial production that contributes economically to the GDP (https://www.intracen.org, accessed on 29 November 2021). Lately, maize production has not met the needs of Thailand, causing more to be imported [3]. On the other hand, rice is a staple food in many countries in the world, including Thailand. Rice contributes immensely to the economy of Thailand. Due to pests and diseases, global warming, and increased drought, maize and rice production are threatened [1]. Until 2016 when it gained entry into West Africa and spread like bush fire, probably due to the favorable climatic conditions, fall armyworm (*Spodoptera frugiperda*) was a native of the Americas [4]. By the year 2018, it had traversed several territories to be reported in India, and once in India, somehow Asia was conquered [4,5]. In late 2018, some of the fields of maize and rice in several provinces of Thailand and Myanmar were already invaded by the fall armyworm, and since then, losses have been incurred [6,7]. This prompted farmers to struggle to control this insect using chemical insecticides, and their efforts have been futile. Eventually, they ended up using a lot of chemicals, which leads to the buildup of the same in the environment and is also likely to enhance the development of resistance by the insect (http://exchange.growasia.org/, accessed on 12 November 2021) [7,8].

Regarding the above, the search for an effective biological control agent to complement the existing chemical control began in the early 1980s after discovering that most of the insects being managed were developing resistance, and also the chemicals had detrimental residues in plants. Additionally, the environment had been impacted negatively [8]. Today, researchers have been able to identify over 750 species of entomopathogenic microorganisms, including fungi, nematodes, bacteria, and viruses, drawn from about 85 genera [8,9,10]. Interestingly, of all the entomopathogens, only fungi do not require ingestion or the openings like nematodes to gain entry into their host, but instead use enzymes to penetrate through the cuticle [11]. Among the identified species of fungi, the most utilized are in the genera *Beauveria*, *Cordyceps*, *Isaria*, *Lecanicillium*, *Metarhizium*, and *Nomuraea,* among others, of which several bio-insecticides are attributed [12,13,14]. Due to their ubiquitous nature and ability to survive in different kinds of environments, these entomopathogenic fungi have been tried against insects in several parts of the world and have emerged victorious in many aspects [12,15,16]. Admittedly, most of them can overwinter in the soil, waiting to infect the next generation of insect hosts, which makes it a perfect biological management choice that is eventually less costly [17]. Several entomopathogenic fungi have been used to successfully control economically important insects in many countries in the world. Reportedly, Thailand has been at the forefront in the isolation and utilization of entomopathogens [11]. When fall armyworm was first reported in Thailand and the neighboring countries, a regional action plan known as ASEAN FAW was set up to find a solution to this destructive pest (https://www.aseanfawaction.org, accessed on 29 November 2021). However, since the entry of fall armyworm in the territory, there is no report of entomopathogenic fungi suggested to manage it. These fungi gain entry into the insect through the cuticle by the use of secreted enzymes or glycosyl transferase-like protein to penetrate through the cuticle. In addition, this protein aids in causing virulence to the insect, leading to mycosis. Research has determined that the *GAS1* gene is responsible for the synthesis of this protein [18,19,20]. Therefore, this study aimed to isolate native entomopathogenic fungi in northern Thailand and investigate their pathogenic ability against the invasive fall armyworm. The obtained fungi were identified through morphological characteristics and multi-gene phylogenetic analyses. Moreover, the *GAS1* gene involved in fungal virulence and aiding the penetration of the insect cuticle was investigated.

## 2. Materials and Methods

### 2.1. Collection and Isolation of Entomopathogenic Fungi

A survey was conducted in Chiang Mai and Lampang provinces of northern Thailand. During the field collection and identification of entomopathogens, the mycosis that is displayed due to the colonization of the insect after the emergence of the hyphae through the cuticle to cause white muscardine disease was employed. The infected cadavers were then taken to the laboratory and isolation of the fungus was performed. The isolation of the native entomopathogens from the insect cadavers and soil samples was conducted using the bait method according to the protocol proposed by Meyling [21]. Pure cultures were kept in the Insect Pathology Laboratory, Department of Entomology and Plant Pathology, Faculty of Agriculture, Chiang Mai University, for further studies.

### 2.2. Identification of Entomopathogenic Fungi

#### 2.2.1. Morphological Study

Morphological observations were performed using pure fungal cultures on potato dextrose agar (PDA) for two weeks at 25 ± 1 °C with a photoperiod of 12:12 h (Dark:Light). Macroscopic observations were determined through colony characteristics obtained by the shape, elevation, growth pattern, color, and texture [9,22,23]. Using a sterile needle, fungal structures were picked up and mounted in lactic acid on glass slides and observed using a compound microscope (ZEISS AX10) by the program ZEN 3.3 (blue edition) to observe hyphal conidiophore and measure conidial sizes. For each isolate, the sizes of 20 conidia were measured following the methods described in previous studies [22].

#### 2.2.2. Molecular Study

Genomic DNA was extracted from fresh fungal mycelium grown on PDA at 28 °C for one week using the DNA Extraction Mini Kit (FAVOGEN, Ping-Tung, Taiwan). Four gene loci: the internal transcribed spacer (ITS), the translation elongation factor 1-alpha (*TEF*-*1*), the RNA polymerase II largest subunit (*RPB1*) and the partial RNA polymerase second largest subunit (*RPB2*) were amplified using polymerase chain reaction (PCR), with ITS4/ITS5, 983F/2218R, RPB1-Af/RPB1-Cr, and fRPB2-5F2/fRPB2-7cR primers, respectively (Table 1). The PCR amplification conditions for ITS and *TEF*-1 are: initial denaturation at 94 °C for 3 min, with 30 cycles of denaturation at 94 °C for 30 s, annealing at 52 °C for 30 s and extension at 72 °C for 45 s, and final extension at 72 °C for 5 min. In addition, PCR conditions for *RPB1* and *RPB2* amplifications are: initial denaturation at 94 °C for 2 min, with 35 cycles of denaturation at 94 °C for 30 s, annealing at 52 °C for 60 s and extension at 72 °C for 1 min, and final extension at 72 °C for 5 min. PCR products were purified using Gel Extraction NucleoSpin^®^ Gel and the PCR Clean-up Kit (Macerey-Nagel, Dueren, Germany). PCR products were sent to commercial sequencing at 1st BASE Company (Kembangan, Malaysia).

The related taxa were identified using the BLAST search (http://blast.ddbj.nig.ac.jp/top-e.html, accessed on 2 November 2021) results and previous publications. Details of the sequences used for the phylogenetic analyses are provided in Table 2. Preliminarily, individual DNA sequence matrixes were aligned by MUSCLE [29] and improved where necessary using BioEdit v.6.0.7 [30]. A phylogenetic tree was constructed under maximum likelihood (ML) and Bayesian inference (BI) methods. The ML analysis was carried out using RAxML-HPC2 on XSEDE (8.2.10) in CIPRES Science Gateway V.3.3 [31] using the GTRCAT model with 25 categories and 1000 bootstrap (BS) replications. The optimum nucleotide substitution model was obtained using the jModel test v.2.3 [32] under the Akaike information criterion (AIC) method. The BI analysis was performed using MrBayes 3.2.6 software for Windows [33]. The selected optimal model of each gene is similar to the GTR+I+G model. Six simultaneous Markov chains were run with one million generations, starting from random trees and keeping one tree every 100th generation until the average standard deviation of split frequencies was below 0.01. The value of burn-in was set to discard 25% of trees when calculating the posterior probabilities. Bayesian posterior probabilities (PP) were obtained from the 50% majority-rule consensus of the trees kept. The tree topologies were visualized in FigTree v1.4.0 [34].

### 2.3. Efficacy Test

#### 2.3.1. Insect Source and Rearing

*Spodoptera frugiperda* used in this study were collected from the demonstration field of the Faculty of Agriculture and Mae-Hia Agricultural Training and Research Center, Chiang Mai University, during the maize growing season. The collected adults and larvae were maintained at the Insect Pathology Laboratory, Department of Entomology and Plant Pathology, Faculty of Agriculture, Chiang Mai University, Thailand. Disease-free neonate larvae were placed in plastic containers (19 cm in width by 27 cm in length by 8 cm in height) and fed on baby corn under laboratory room conditions at 26 ± 1 °C, with a photoperiod of 12:12 h (Dark:Light) and 65% ± 5% humidity [17]. The egg batches oviposited were maintained under controlled laboratory conditions, and upon hatching, the neonate larvae were individually placed in plastic containers measuring 6 × 3 cm, and 3 oz.

#### 2.3.2. Fungal Inoculum Preparation

For each isolate, stock suspension was prepared in 250 mL reagent bottles with the addition of the mass sporulating culture in 100 mL of distilled water, with 0.01% Tween 80. Thereafter, the surface was softly scraped to dislodge the conidia using a sterile loop [9]. The suspensions were pipetted from the plates. The mixture was then vigorously shaken for 3 min and then filtered. The hyphae were isolated through a sterilized millipore cloth after blending. An improved Neubauer hemocytometer was used to determine conidial concentration under a light microscope at 400× magnification [22]. After this, two different concentrations, 1 × 10^6^ and 1 × 10^8^ mL^−1^ conidia, were obtained by dissolving the original solution in sterilized distilled water with 0.01% Tween 80 [46]. These concentrations were used for the efficacy bioassay [13].

#### 2.3.3. Insect Bioassay

Two days after hatching, the second instar larvae were dipped into the two different fungal concentrations (1 × 10^6^ and 1 × 10^8^ mL^−1^ conidia) obtained in the process described above and returned into the plastic container supplied with about 5 g of baby corn. For the control, sterilized distilled water with 0.01% Tween 80 was used. This process was repeated three times after every three days, during which the food was also changed [47]. Mortality data were recorded every three days. The experiment followed a completely randomized block design with three replications for each concentration. Each replicate had 30 larvae. The experiments were independently repeated twice. Mortality data were analyzed using one-way ANOVA and presented as a percentage, as indicated in Table 4. The treatment means were compared using Tukey’s test for their significance at the 0.05% probability level. The mortalities were compared by the F-test. Differences were considered significant at *p* < 0.05. The IBM SPSS Statistical Software package version 23.0 (IBM Corp., Armonk, NY, USA, 2015) was used to conduct the statistical analyses.

### 2.4. Molecular Characterization of GAS1 Gene

Fungal DNA of each isolate was extracted from mycelia covered on fall armyworm by the DNA Extraction Mini Kit (FAVOGEN, Ping-Tung, Taiwan) following the manufacturer’s protocol. The *GAS1* gene was amplified using the specific primers GTF2 (5′-CCCGTCA TCTCCTTGCTCATCAG-3′) and GTR2 (5′-GTCATCAACGAAAAGGGCAACGAG-3′), following the study of Zhang et al. [19]. The PCR amplification conditions for *GAS1* followed the initial denaturation at 95 °C for 3 min, with 35 cycles of denaturation at 95 °C for 30 s, annealing at 56 °C for 30 s and extension at 72 °C for 1 min, and final extension at 72 °C for 10 min. PCR products were purified using Gel Extraction NucleoSpin^®^ Gel and the PCR Clean-up Kit (Macerey-Nagel, Dueren, Germany). PCR products were sent to commercial sequencing at 1st BASE Company (Kembangan, Malaysia). Purified PCR products were sequenced using a commercial provider obtained from the 1st BASE Company (Kembangan, Malaysia). The sequences were assembled and then subjected to BLAST search in the GenBank database (http://blast.ddbj.nig.ac.jp/top-e.html, accessed on 10 November 2021).

## 3. Results

### 3.1. Collection and Morphological Characterization of Entomopathogenic Fungi

After carrying out preliminary observations on the isolates collected during the field survey, six fungal isolates were obtained, namely BCMU1, BCMU2, BCMU3, BCMU4, BCMU5, and BCMU6, as shown in Table 2. Morphologically, all six isolates examined were typical of the *Beauveria* genus due to the features displayed. Microscopic and macroscopic observations of the morphology confirmed the typical characteristics of *B. bassiana,* displaying white to the yellowish coloration on the mycelium on the PDA media (Figure 1 and Appendix A).

We observed conidiogenous cells that are short and globose. Additionally, the conidiophores had whorls and clustered compactly. The mycelium was observed to be cottony and closely appressed to the media, with all of the isolates white on the top side except for BCMU4, which was off-white, while on the underside, most of them displayed a yellowish coloration save for BCMU5, which was brownish in appearance. In terms of shape, four of the isolates were round in shape while two were almost oval. They were all raised in terms of elevation and smooth in texture. Three of them had a dispersed growth pattern while three were dense and disperse. However, fungal identification was confirmed by multi-gene phylogenetic analyses.

### 3.2. Phylogenetic Results of Obtained Entomopathogenic Fungi

The sequences of six fungal strains in this study were deposited in the GenBank database (Table 2). The alignment of a combination of ITS, *TEF-1*, *RPB1*, and *RPB2* genes contained 3185 characters, including gaps (ITS: 1–563, *TEF-1*: 564–1501, *RPB1*: 1502–2257, and *RPB**2*: 2258–3185). RAxML analysis of the combined dataset yielded a best-scoring tree with a final ML optimization likelihood value of −10,287.872. The matrix contained 622 distinct alignment patterns with 8.74% undetermined characters or gaps. Estimated base frequencies were recorded as follows: A = 0.2431, C = 0.2867, G = 0.521, T = 0.2179, and substitution rates of: AC = 1.3261, AG = 4.2754, AT = 1.2198, CG = 0.8715, CT = 10.6195, and GT = 1.0000. The gamma distribution shape parameter alpha was equal to 0.2350 and the Tree-Length was equal to 0.4903. In addition, the final average standard deviation of the split frequencies at the end of the total MCMC generations was calculated as 0.00837 through BI analysis. Phylograms of the ML and BI analyses were similar in terms of topology (data not shown). Therefore, the phylogram obtained from the ML analysis was selected and presented for this study. The phylogram was comprised of 34 sequences of *Beauveria* strains and two sequences (*Isaria farinosa* ARSEF 4029 and *Lecanicillium antillanum* CBS 350.85) of the outgroup (Figure 2). Our analysis confirmed that all fungal strains in this study (BCMU1, BCMU2, BCMU3, BCMU4, BCMU5, and BCMU6) belonged to the *B*. *bassiana,* with high support values (BS = 100% and PP = 1.0). *Beauveria bassiana* formed the sister clade to *B*. *kipukae*, *B*. *lii,* and *B*. *varroae,* with high support (BS = 98% and PP = 1.0).

### 3.3. Efficacy Test

Generally, all six isolates caused mortality to fall armyworm to a greater extent, as displayed in Table 3. Apparently, there were significant differences in their efficacy, with the most efficacious isolate being BCMU6 at a concentration of 1 × 10^8^ mL^−^^1^. Interestingly, by the third day, this isolate had caused about 43% mortality to the larvae of *S*. *frugiperda,* which was an impressive outcome. After twelve days of observation, the same isolate had caused mortality of 91.67%, while the least mortality observed was on the isolate BCMU1 in both concentrations of 1 × 10^6^ and 1 × 10^8^ mL^−1^, respectively.

### 3.4. Molecular Characterization of GAS1 Gene

Molecular characterization of the *GAS1* gene from fungal genomic DNA was performed by PCR amplification. The result showed that the *GAS1* gene obtained from each fungal isolate showed 100% similarity to *B. bassiana* ARSEF2860 (Table 4). Additionally, the obtained *GAS1* gene showed 89.31% and 88.12% similarity to *Cordyceps militaris* ATCC 34164 and *Isaria fumosorosea* ARSEF2679, respectively. Therefore, this study confirms the presence of the *GAS1* gene from obtained fungi.

## 4. Discussion

This study sought to molecularly and morphologically characterize six isolates of *Beauveria bassiana* that were previously isolated from insect cadavers, and also to determine their efficacy against the invasive *Spodoptera frugiperda.* The search for biological control against fall armyworm intensifies by the day in order to avoid the development of resistance to the chemical control measures that have been observed in its native region [48]. The morphology of the six isolates studied was similar to *B. bassiana,* as described in previous studies [46,49]. For a long time, morphological analysis had been the traditional tool used in the characterization of various entomopathogenic fungi, until it was discovered that the conidial features can easily change in the process of culturing. For example, *B. bassiana* and *B. asiastica* have been observed to be morphologically similar. Therefore, the use of multi-gene molecular analysis comes in handy as a confirmatory tool [50]. Since the year 1990, molecular work has been put into action for the identification of entomopathogenic fungi, especially of the genus *Beauveria* [51].

*Beauveria bassiana* has been in the limelight for a long time as far as efficacy is concerned. In the present study, all the isolates used in the study caused mortality, but at differing rates, with the highest instigating up to 91.67% after twelve days since the initial inoculation. The second highest isolate caused 83.33% mortality after twelve days. Previous studies have recorded mortalities of 100%, which is incredible, but mortalities of 60% and above are good enough in controlling insect populations [52]. *Beauveria*
*bassiana* has been an entomopathogen in many insect orders, including Lepidoptera, Hemiptera, Coleoptera, and Diptera, among others. Particularly, Lepidopteran insects have been successfully controlled by this fungus, as can be ascertained by the study performed on *Plutella xylostella,* where it caused over 74% mortality [53]. Additionally, when inoculated against *Galleria mellonella*, *B*. *bassiana* caused mortalities within a short time, which was a promising gesture toward the management of this particular insect [54]. Further, it has been previously confirmed that the lethal action of *B*. *bassiana* is proportionally dependent on the concentration. The higher the concentration, the more efficacious it is to the insect [55]. Looking at our results, this fact is corroborated. In all the isolates studied, the highest mortalities were achieved with concentration of 1 × 10^8^ mL^−1^. However, this is not a cause for alarm as these regimens are safer both for the environment and for animals, including humans [55]. Although our study was laboratory-based, we investigated some records on the endophytic influence that may be caused by *B*. *bassiana* on plants, and it is clear that it is friendly and does not cause any significant changes in the plant fresh and dry weight as well as the nutrient elements [56]. In fact, the endophytic strains have been reported to promote plant growth and at the same time help in controlling insect pests [57,58].

Finally, the *GAS1* gene was found to be present in the six isolates of *B. bassiana* upon analysis. This points to the fact that the virulence caused against the fall armyworm is aided by the presence of this gene [19]. Xiao et al. observed that *B.*
*bassiana* has many species-specific virulence genes, with *GAS*1 being among them [59]. Lai et al. [60] also observed the upregulation of the *BBGAS1* gene prior to *B*. *bassiana* gaining entry into the hemocoel of the *Anopheles* mosquito. These records further corroborate the role the *GAS*1 gene plays in the infection and virulence process. Additionally, the *GAS1* gene is known to code for conidial thermotolerance [20]. To further prove that the *GAS1* gene is key to the infection process, a study carried out by Cao et al. in 2012 resulted in reduced cuticle penetration when they deleted the *Magas1* gene in *Metarhizium acridum* [20]. Therefore, our findings are proof that these isolates of *B*. *bassiana* possess an important gene that is a key contributor to the insect infection process and virulence. Furthermore, the conidial thermotolerance role that the gene plays is valuable when *B*. *bassiana*-based commercial biopesticides are used in an extreme environment. In addition, *B. bassiana* has been reported to produce secondary metabolites, such as beauvericin, tenellin, oosperein, bassianin, and bassianolide, which are capable of enervating the immune systems of insect hosts [61,62].

## 5. Conclusions

We confirmed that our indigenous fungal isolates are *B. bassiana* through morphological observations as well as multi-gene molecular analyses. This is the first report of *B. bassiana* instigating efficacy against *S. frugiperda* in Thailand, which contributes valuable knowledge towards the search for an eco-friendly solution to the invasive insect. Our data will be helpful in the identification and characterization of *B. bassiana* isolates that are ubiquitous and effective in the control of several invasive insects around the world. Additionally, the finding that one of the isolates could cause mortality of up to 91.67% to the fall armyworm is a great inspiration towards the continuous search for a biological solution to fall armyworm. Notwithstanding, the *GAS1* gene, which is known to code for cuticle penetration, conidial thermotolerance, and virulence against insect hosts, was found to be present in all the isolates investigated. Moreover, our research provides a pedestal for similar studies in the future. However, we recommend further research to be performed, including field experiments to determine whether there are effects attributed to ultraviolet rays and any other biotic and abiotic factors. We hope that this will lead to the development of a strategy for the commercialization of an indigenous *B. bassiana*-based bio-insecticide against *S. frugiperda.*

## Figures and Tables

**Figure 1 jof-07-01073-f001:**
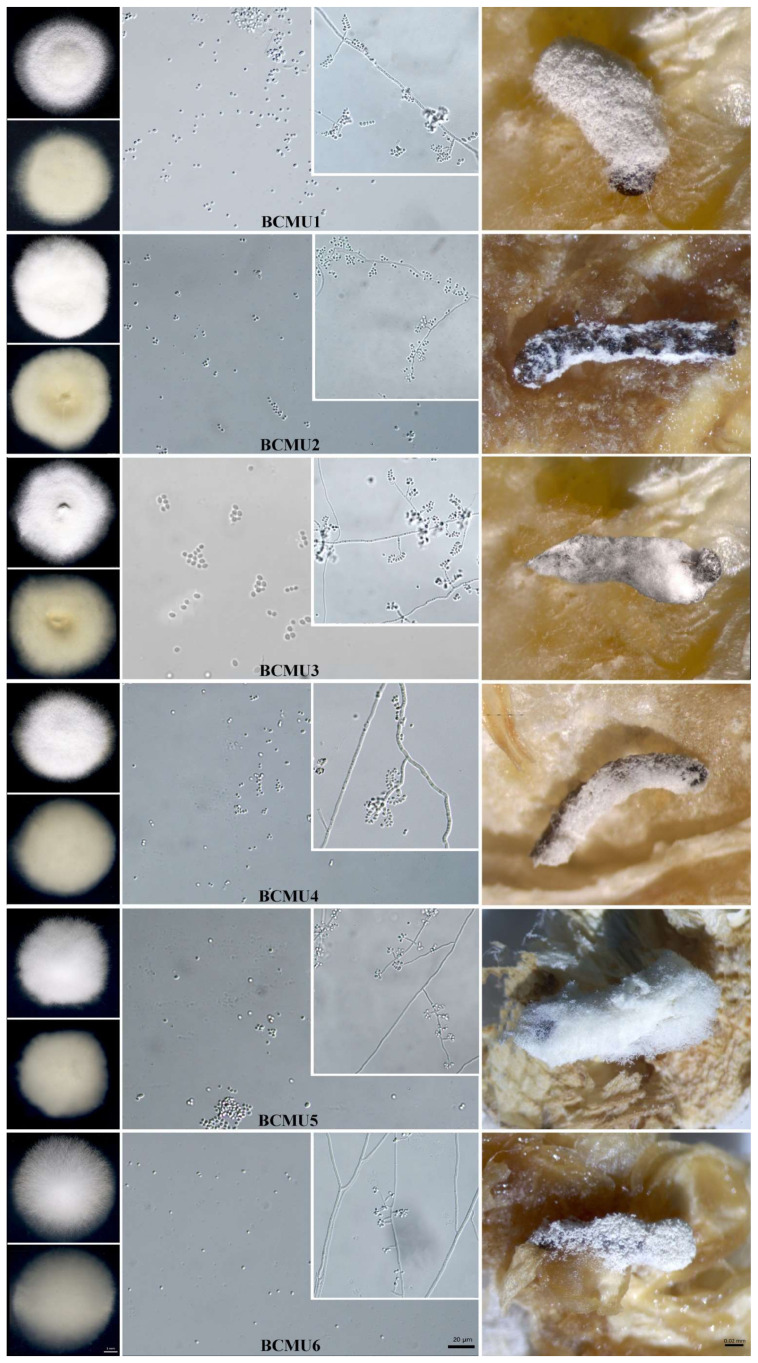
Pictorial presentation of BCMU1–BCMU6 colony on the obverse and reverse sides on PDA media, conidia, and the hyphae and the mycosis caused on *Spodoptera frugiperda*. Scale Bar = 1 µm, 20 µm, and 0.02 mm respectively. The isolates were cultured on potato dextrose agar for 14 days at 25 ± 1 °C with a photoperiod of 12:12 h (Dark: Light). Once the larvae had died, they were placed on moist conditions to allow mycosis.

**Figure 2 jof-07-01073-f002:**
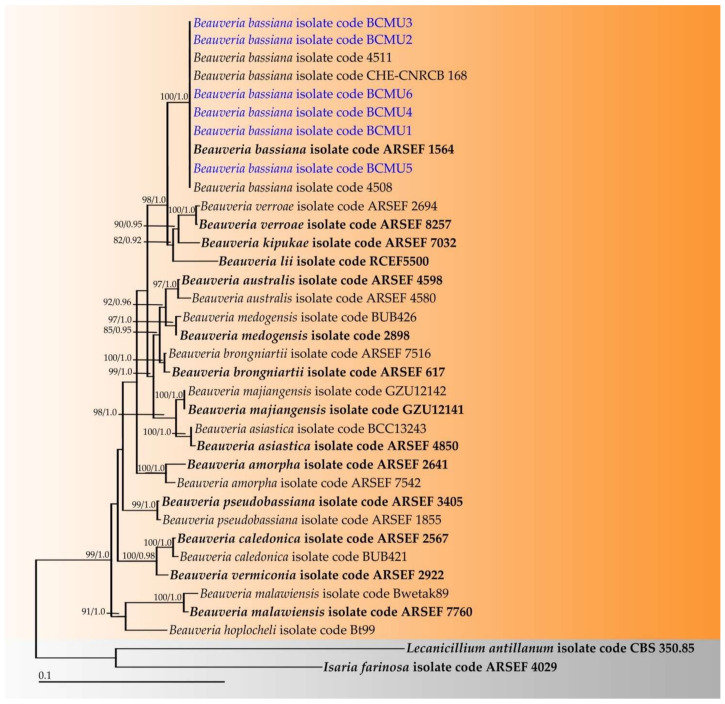
Phylogram derived from maximum likelihood analysis of 36 sequences of the combined ITS, *TEF-1*, *RPB1*, and *RPB2* sequences. *Isaria farinosa* ARSEF 4029 and *Lecanicillium antillanum* CBS 350.85 were used as the outgroup. The numbers above branches represent bootstrap percentages (**left**) and Bayesian posterior probabilities (**right**). Bootstrap values > 75% and Bayesian posterior probabilities > 0.90 are shown. The scale bar represents the expected number of nucleotide substitutions per site. Sequences obtained in this study are in blue. Type strains are indicated in bold.

**Table 1 jof-07-01073-t001:** Primers used in this study.

Gene	Primer Name	Primer Sequence	Reference
ITS	ITS1F	CTTGGTCATTTAGAGGAAGTAA	[24]
	ITS4	TCCTCCGCTTATTGATATGC	[25]
*TEF-1*	983F	GCTCCYGGHCAYCGTGAYTTYAT	[26]
	2218R	ATGACACCRACRGCRACRGTYTG	[26]
*RPB1*	RPB1-Af	GAR TGYCCDGGDCAYTTYGG	[27]
	RPB1-Cr	CCNGCDATNTCRTTRTCCATRTA	[27]
*RPB2*	fRPB2-5f2	GAYGAYMGWGATCAYTTYGG	[28]
	fRPB2-7cR	CCCATRGCTTGYTTRCCCAT	[28]

**Table 2 jof-07-01073-t002:** GenBank sequences data of fungal isolates used in this study.

Strain and Voucher No.	Taxon	Country	Host/substratum	GenBank Accession Number				Reference
				ITS	*TEF-1*	*RPB1*	*RPB2*	
BCMU1	*Beauveria bassiana*	Thailand	*Bactocera dorsalis*	OL375165	OL410297	OL410303	OL410309	This study
BCMU2	*Beauveria bassiana*	Thailand	Coffeeberry borer	OL375167	OL410298	OL410304	OL410310	This study
BCMU3	*Beauveria bassiana*	Thailand	Ant	OL375168	OL410299	OL410305	OL410311	This study
BCMU4	*Beauveria bassiana*	Thailand	Coffeeberry borer	OL375169	OL410300	OL410306	OL410312	This study
BCMU5	*Beauveria bassiana*	Thailand	Coffee stem borer	OL375170	OL410301	OL410307	OL410313	This study
BCMU6	*Beauveria bassiana*	Thailand	*Bactocera* *dorsalis*	OL375173	OL410302	OL410308	OL410314	This study
ARSEF 1564 ^T^	*Beauveria bassiana*	Italy	*Hyphantria* *cunea*	GU734762	EF222318	HQ880833	HQ880905	[35]
CHE-CNRCB 168	*Beauveria bassiana*	Mexico	*Diaphorina citri*	KU725691	KU725693	KU725699	KU725703	[36]
Isolate 4511	*Beauveria bassiana*	China	Soil	KX901310	KX901322	KX901328	KY464981	[37]
Isolate 4508	*Beauveria bassiana*	China	Soil	KX901307	KX901319	KX901325	KY464978	[37]
2898 ^T^	*Beauveria medogensis*	China	Soil	KU994837	KU994833	KU994835	KU994834	[37]
BUB426	*Beauveria medogensis*	China	Soil	MG642832	MG642904	MG642859	MG642874	[37]
RCEF5500 ^T^	*Beauveria lii*	China	*Henosepilachna vigintioctopunctata*	JN689372	JN689371	JN689374	JN689370	[38]
ARSEF 8257 ^T^	*Beauveria verroae*	France	*Varroa destructor*	NR111599	HQ881002	HQ880872	HQ880944	[39]
ARSEF 2694	*Beauveria verroae*	Switzerland	*Larinus* sp.	HQ880802	HQ881004	HQ880874	HQ880946	[39]
ARSEF 4598 ^T^	*Beauveria australis*	Australia	Soil	NR111597	HQ880995	HQ880861	HQ880933	[39]
ARSEF 4580	*Beauveria australis*	Australia	Orthoptera: Acrididae	HQ880788	HQ880994	HQ880860	HQ880932	[39]
ARSEF 7032 ^T^	*Beauveria kipukae*	USA	Not provided	NR111600	HQ881005	HQ880875	HQ880947	[39]
ARSEF 7760 ^T^	*Beauveria malawiensis*	Malawi	*Phoracantha* *semipunctata*	DQ376247	DQ376246	HQ880897	HQ880969	[40]
Bwetak89	*Beauveria malawiensis*	New Zealand	Not provided	MW027837	MW030946	MW027830	MW027829	[35]
ARSEF 4850 ^T^	*Beauveria asiastica*	Korea	Coleoptera: Cerambycidae	NR111596	KJ523141	HQ880859	HQ880931	[39]
BCC13243	*Beauveria asiastica*	Thailand	NR	MN401629	MN401455	MN401553	NR	[41]
ARSEF 2567 ^T^	*Beauveria caledonica*	Scotland	Soil	HQ880817	EF469057	HQ880889	HQ880961	[39]
BUB421	*Beauveria caledonica*	China	Soil	MG642831	MG642903	MG642858	MG642873	[39]
GZU12141 ^T^	*Beauveria majiangensis*	China	Coleoptera	MG052642	MG052640	MG052644	NR	[42]
GZU12142	*Beauveria majiangensis*	China	Coleoptera	MG052643	MG052641	MG052645	NR	[42]
ARSEF 617 ^T^	*Beauveria brongniartii*	France	Coleoptera: Scarabaeidae	NR111595	HQ880991	HQ880854	HQ880926	[39]
ARSEF 7516	*Beauveria brongniartii*	Japan	Coleoptera: Scarabaeidae	HQ880766	HQ880976	HQ880838	HQ880910	[39]
Bt99	*Beauveria hoplocheli*	Reunion Island	Coleoptera: Melolonthidae	KC339698	KC339710	KM453949	KM453958	[43]
ARSEF 3405 ^T^	*Beauveria pseudobassiana*	Kentucky, USA	Lepidoptera: Tortricidae	NR111598	NR	HQ880864	HQ880936	[39]
ARSEF 1855	*Beauveria pseudobassiana*	Canada	Coleoptera: Scolytidae	HQ880796	HQ880999	HQ880868	HQ880940	[39]
ARSEF 2922 ^T^	*Beauveria vermiconia*	Chile	Soil	NR151832	NR	HQ880894	HQ880966	[44]
ARSEF 2641 ^T^	*Beauveria amorpha*	Brazil	Hymenoptera: Formicidae	NR111601	NR	HQ880880	HQ880952	[39]
ARSEF 7542	*Beauveria amorpha*	Colorado, USA	Hymenoptera: Formicidae	HQ880805	HQ881007	HQ880877	HQ880949	[39]
CBS 350.85	*Lecanicillium antillanum*	Cuba	Hymenomycete: Agaric	MH861888	DQ522350	DQ522396	DQ522450	[45]
ARSEF 4029	*Isaria farinosa*	Denmark	Coleoptera: Carabidae	HQ880828	HQ881019	HQ880900	HQ880972	[39]

“NR” = Not reported. Superscript “T” indicates type strain

**Table 3 jof-07-01073-t003:** Percentage of cumulative mortalities caused of *S*. *frugiperda* by the six isolates of *B*. *bassiana*.

Isolates	Time (Days)			
	3	6	9	12
BCMU1 10^8^	3.33 ± 1.67 ab*	5.00 ± 0.00 a	6.67 ± 1.67 a	41.67 ± 3.33 c
BCMU1 10^6^	3.33 ± 3.33 ab	3.33 ± 3.33 a	3.33 ± 3.33 a	10.00 ± 2.89 a
BCMU2 10^8^	15.00 ± 0.00 bc	20.00 ± 2.87 b	31.67 ± 3.33 cd	41.67 ± 3.33 c
BCMU2 10^6^	0.00 ± 0.00 a	3.33 ± 3.33 a	5.00 ± 2.88 a	35.00 ± 2.89 bc
BCMU3 10^8^	36.67 ± 3.33 ef	40.00 ± 2.87 c	43.33 ± 3.33 e	73.33 ± 3.33 ef
BCMU3 10^6^	3.33 ± 1.67 ab	11.67 ± 1.67 ab	11.67 ± 1.67 ab	55.00 ± 2.89 d
BCMU4 10^8^	30.00 ± 2.89 de	53.33 ± 3.33 d	55.00 ± 2.87 f	60.00 ± 2.89 d
BCMU4 10^6^	21.67 ± 1.67 cd	21.67 ± 1.67 b	21.67 ± 1.67 bc	23.33 ± 1.67 b
BCMU5 10^8^	28.33 ± 4.41 de	56.67 ± 7.26 d	66.67 ± 4.41 g	83.33 ± 6.01 fg
BCMU5 10^6^	3.33 ± 1.67 ab	8.33 ± 1.67 ab	8.33 ± 1.67 a	35.00 ± 0.00 bc
BCMU6 10^8^	43.33 ± 6.00 f	71.67 ± 4.41 e	76.67 ± 3.33 g	91.67 ± 1.67 g
BCMU6 10^6^	15.00 ± 2.87 bc	23.33 ± 3.33 b	33.33 ± 1.67 de	61.67 ± 3.33 de
Control	0.00 a	0.00 a	0.00 a	0.00 a
df1	12	12	12	12
df2	26	26	26	26
F-test 0.05	27.798	50.594	91.074	81.081

* The lower case letters a, b and c show significant differences in mortalities caused by the different concentration of the isolates.

**Table 4 jof-07-01073-t004:** Details of the *GAS1* sequences obtained from entomopathogenic fungi in this study.

Fungal Isolate	Length (bp)	GenBank Accession Number	Closeted Species/Accession Number	Similarity (%)
BCMU1	346	OL469003	*Beauveria bassiana* ARSEF 2860/XM008599737	100
BCMU2	358	OL469004	*Beauveria bassiana* ARSEF 2860/XM008599737	100
BCMU3	356	OL469005	*Beauveria bassiana* ARSEF 2860/XM008599737	100
BCMU4	340	OL469006	*Beauveria bassiana* ARSEF 2860/XM008599737	100
BCMU5	346	OL469007	*Beauveria bassiana* ARSEF 2860/XM008599737	100
BCMU6	350	OL469008	*Beauveria bassiana* ARSEF 2860/XM008599737	100

## Data Availability

The DNA sequences data obtained in this study have been deposited in GenBank under accession numbers ITS (OL375165, OL375167, OL375168, OL375169, OL375170, OL375173, GU734762), *TEF*-1 (OL410297, OL410298, OL410299, OL410300, OL410301, OL410302), *RPB*1 (OL410303, OL410304, OL410305, OL410306 OL410307 OL410308), *RPB*2 (OL410309, OL410310, OL410311, OL410312, OL410313, OL410314), and *GAS1* (OL469003, OL469004, OL469005, OL469006, OL469007, OL469008).

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
