# Peer review of "Evaluation of Native Entomopathogenic Fungi for the Control of Fall Armyworm (Spodoptera frugiperda) in Thailand: A Sustainable Way for Eco-Friendly Agriculture"

_jof, 2021, doi:10.3390/jof7121073_

Round 1

Reviewer 1 Report

The authors of the manuscript entitled “Evaluation of native entomopathogenic fungi for the control of fall armyworm (Spodoptera frugiperda) in Thailand: a sustainable way for eco-friendly agriculture” conducted the field survey to investigate entomopathogenic fungi resources in Thailand, and combined morphological and molecular methods to identify them and finally evaluate their potential on controlling S. frugiperda larvae in laboratory assay. The highest mortality was achieved up to 91.67% with isolate BCMU6 at the higher concentration. The study is very interesting and of great importance for improving biological control and reduction of using chemical pesticides on fall armyworm. The results could provide valuable suggestions for the use of entomopathogenic fungi in practice. I recommend to accept it for publication after minor revision.

In this manuscript the authors reported six isolates belonged to Beauveria bassiana, I wonder if any other entomopathogenic fungi were found in the field survey, such as Metarrhizium anisopliae?

Can you tell us the reason for choosing these two fungal concentration (1×106 and 1×108 conidia mL-1) in insect bioassay? Whether the control efficacy (e.g., mortality) could be consolidated or accelerated by increasing the conidia concentration?

Did you compare the efficacy between different methods of applying biological agents e.g., dipping or spraying on the leaf? Spraying will be more feasible in practice.

The authors showed the isolate BCMU6 had well effects on controlling the second instar larvae. How about the third and higher instar larvae? If the isolate BCMU6 can not kill S. frugiperda in an early stage, they will enter to higher instar larvae stage quickly, becoming more resistant to entomopathogenic fungi.

Author Response

#

ISSUE RAISED

RESPONSE

1.      

In this manuscript the authors reported six isolates belonged to Beauveria bassiana, I wonder if any other entomopathogenic fungi were found in the field survey, such as Metarrhizium anisopliae?

Yes, the field survey realized several entomopathogens and the ubiquitous Metarhizium anisopliae was among them and others such as Isaria spp. among others. However, in this study, we focused on the Beauveria bassiana.

2.      

Can you tell us the reason for choosing these two fungal concentration (1×10and 1×108conidia mL-1) in insect bioassay? Whether the control efficacy (e.g., mortality) could be consolidated or accelerated by increasing the conidia concentration?

Before choosing these concentrations, we dug through the literature and a huge amount of data revealed that 1x108conidia mL-1 gives the highest efficacy. We also carried out preliminary tests and 1x108 mL-1 was efficient comparatively. In this experiment, we used the two for comparison purposes. Arguably, increasing conidial concentration my lead to high mortality within a short time, however, we also consider the cost of production.

3.      

Did you compare the efficacy between different methods of applying biological agents e.g., dipping or spraying on the leaf? Spraying will be more feasible in practice.

We compared both spraying and dipping but dipping was more effective and convenient in the laboratory. Spraying is effective and applicable in the field and we are currently doing a study in the greenhouse to affirm this. 

4.      

The authors showed the isolate BCMU6 had well effects on controlling the second instar larvae. How about the third and higher instar larvae? If the isolate BCMU6 can not kill S. frugiperda in an early stage, they will enter to higher instar larvae stage quickly, becoming more resistant to entomopathogenic fungi.

Observably, it is advisable to spray the first and second instar larval stages of S. frugiperda as they respond better. As they advance, they tend to molt rapidly hence jeopardizing the effect of the EPF.  

Reviewer 2 Report

The manuscript addresses an important subject, the biological control of fall armyworm. This is an area of interest for many researchers faced with the fall armyworm problem. I feel there are areas that need to be improved for clarity.

  1. The English can be improved further.
  2. I would suggest used of efficacy instead of entomopathogenicity
  3. Abstract: Lines 34 and 35 -  split the sentence. Writing on lines 40 to 42 can be made more focused.

Introduction

  1. Needs to be revised to flow properly
  2. State the actual contribution of maize to GDP
  3. Lines 67 - 86, describe what is known about EPF and their use to control FAW. Also, highlight the gaps in the case of Thailand

Methods

  1. Justify the use of two concentrations
  2. State what control was used. It only appears in the results

Results

  1. First part of 3.1 (lines 199 to 202) should go to the methods section
  2. Look for a better word to replace instigated

Overall, the them of the paper should be to look for effective biocontrol agents as complementary, and not necessarily a replacement for synthetic insecticides.

If the isolates where already in stock, and were just evaluated. This should be made clear, and the other sections that describe earlier handling omitted

Author Response

#

ISSUE RAISED

RESPONSE

  1.  

The English can be improved further.

Done.

2.      

I would suggest used of efficacy instead of entomopathogenicity

The word entomopathogenicityhas been replaced by efficacy.

3.

Abstract: Lines 34 and 35 -  split the sentence. Writing on lines 40 to 42 can be made more focused.

The sentence has been split and the lines 40 to 42 made more focused by narrowing to Beauveria bassiana.

4.

Introduction needs to be revised to flow properly

Done.

5.

State the actual contribution of maize to GDP

It is not indicated explicitly the contribution of maize to the GDP but the agricultural sector contributes about 9.9% to the Thailand GDP and maize is a key crop in the same (https://www.intracen.org). This has been included in the text.

6.

Lines 67 - 86, describe what is known about EPF and their use to control FAW. Also, highlight the gaps in the case of Thailand

The gaps have been highlighted. Fortunately, Thailand has a huge collection of native entomopathogens and some continue to be discovered by the day. However, the EPF for the control of FAW has not been determined but we assume that the scientists are working on the same and we will be seeing more to come. In addition, there is an organization in place known as ASEAN FAW https://www.aseanfawaction.orgfocusing on Fall armyworm and how to control it in Southeast Asia suggesting how serious this pest is.

7.

Justify the use of two concentrations

Before choosing these concentrations, we dug through the literature and a huge amount of data revealed that 1x108conidia mL-1 gives the highest efficacy. We also carried out preliminary tests and 1x108 mL-1 was efficient comparatively. In this experiment, we used the two for comparison purposes. Arguably, increasing conidial concentration my lead to high mortality within a short time, however, we also consider the cost of production.

8.

State what control was used. It only appears in the results

Sterilized distilled water with 0.01% Tween 80 was the control used. It is now stated in the materials and methods.  

9.

First part of 3.1 (lines 199 to 202) should go to the methods section

The paragraph has been moved to materials and methods section.

10.

Look for a better word to replace instigated

Instigated has been replaced with the word caused